# What Is in Your Shark Fin Soup? Probably an Endangered Shark Species and a Bit of Mercury

**DOI:** 10.3390/ani12070802

**Published:** 2022-03-22

**Authors:** Christina Pei Pei Choy, Benjamin J. Wainwright

**Affiliations:** 1Independent Researcher, Singapore 521112, Singapore; christinachoy01@gmail.com; 2Yale-NUS College, National University of Singapore, Singapore 138527, Singapore

**Keywords:** CITES, IUCN, conservation, Singapore, DNA barcoding, mercury

## Abstract

**Simple Summary:**

Shark fin soup is consumed by many Asian communities throughout the world and is one of the main drivers of the demand for shark fin. The demand for shark products has seen shark populations decline by as much as 70%. The fins found in soups break down into a fibrous mass meaning that identifying the species of shark that a fin came from is impossible by visual methods. Here, we use molecular techniques to identify the species of sharks found in bowls of soup collected in Singapore. We identified a number of endangered species in the surveyed soups, and many of these species have been shown to contain high levels of mercury, a potent neurotoxin. It is highly likely that consumers of shark fin soup are consuming levels of mercury that are above safe allowable limits, and at the same time are contributing to the massive declines in global shark populations.

**Abstract:**

Shark fin soup, consumed by Asian communities throughout the world, is one of the principal drivers of the demand of shark fins. This near USD 1 billion global industry has contributed to a shark population declines of up to 70%. In an effort to arrest these declines, the trade in several species of sharks is regulated under the auspices of the Convention on International Trade in Endangered Species of Wild Fauna and Flora (CITES). Despite this legal framework, the dried fins of trade-regulated sharks are frequently sold in markets and consumed in shark fin soup. Shark fins found in soups break down into a fibrous mass of ceratotrichia, meaning that identifying the species of sharks in the soup becomes impossible by visual methods. In this paper, we use DNA barcoding to identify the species of sharks found in bowls of shark fin soup collected in Singapore. The most common species identified in our samples was the blue shark (*Prionace glauca*), a species listed as Near Threatened on the International Union for Conservation of Nature (IUCN) Red List with a decreasing population, on which scientific data suggests catch limits should be imposed. We identified four other shark species that are listed on CITES Appendix II, and in total ten species that are assessed as Critically Endangered, Endangered or Vulnerable under the IUCN Red List of Threatened Species. Globally, the blue shark has been shown to contain levels of mercury that frequently exceed safe dose limits. Given the prevalence of this species in the examined soups and the global nature of the fin trade, it is extremely likely that consumers of shark fin soup will be exposed to unsafe levels of this neurotoxin.

## 1. Introduction

Shark fin soup is considered a delicacy served in many Asian communities throughout the world [1,2]; it is also highly prized in traditional Chinese Medicine [3,4,5] where it is thought to help to alleviate a host of ailments and have beneficial properties throughout the body. The fishing industry that supplies the fins for this dish is arguably one of the principle drivers of shark overexploitation [6,7], with declines of 71% reported for oceanic sharks since the 1970s [8]. These declines are largely attributed to the increased fishing efforts required to meet growing market demands, with predictions suggesting that shark consumption and further declines will accelerate if regulations are not effectively enforced [7,8]. This overexploitation has led to the inclusion of several shark species under the Convention on International Trade in Endangered Species of Wild Fauna and Flora (CITES) Appendix II. This list includes species not necessarily threatened with extinction, but in which trade must be controlled in order to avoid utilization incompatible with their survival (CITES, 1994). Unfortunately, work from around the globe regularly shows that fins from CITES-listed sharks are traded within, and throughout many countries [7,9,10]. This trade is made possible and is extremely difficult to prevent because once a fin is removed from a carcass and processed for sale, the majority of the diagnostic features that can be used in visual identification are lost, then becoming nearly impossible to identify the species of shark to which a fin belonged without molecular methods [9,11,12]. The practice of finning occurs at sea; the high value fins are removed and the lower value carcass is discarded at sea [13]. This maximizes the number of fins that can be collected and minimizes the storage space occupied by the lower value carcass.

The consumption of seafood can lead to the inadvertent ingestion of mercury by humans where it is associated with a number of adverse health risks [14]. Mercury (Hg) poisoning has been implicated in central nervous system and brain damage, infant death, and can retard fetal cognitive development when mothers consume mercury-containing seafood [15,16]. The burning of fossil fuels is the primary source of atmospheric mercury [17], which then dissolves in the oceans, where Hg concentrations in surface waters have increased by a factor of three since the industrial revolution [18,19,20]. In marine systems, bacteria transform mercury into its organic form, methylmercury (MeHg), where it has the potential to accumulate and biomagnify in large upper trophic level predators, such as sharks [20,21,22,23]. Unsurprisingly, shark fins consumed in soups frequently exceed safe mercury concentrations, with some studies showing all examined samples significantly above the established maximum limits for mercury consumption [22].

Shark fin soup is primarily made up of collagenous protein fibers, or ceratotrichia that are found on the inside of the fin. The consumption of this dish and other shark products offers a potential pathway that can expose humans to elevated levels of mercury. Studies show that children and adults who consume shark products once a week are exposed to three times more mercury than what is recommended by the United States Environmental Protection Agency (U.S. EPA) [24]. While the majority of studies generally report mercury and levels of other toxic elements in muscle tissue, the liver or from vertebrae where they readily accumulate [20,24,25], it is important to note that fins can contain higher levels of mercury and other toxic elements, such as lead and cadmium, in comparison to muscle and other non-fin tissues [23,26]. Additionally, work performed in Hong Kong and China shows that the total amount of mercury found in shark fins regularly exceeded the prescribed Hong Kong and China legal limit of 0.5 ppm [22], and the 1 ppm legal limit used for predatory fish in Singapore [27].

Different species of sharks contain varying levels of toxic elements; for example, the mean percentage of mercury found in silky sharks is nearly twice that found in the scalloped hammerhead, and there is a general trend of higher levels of mercury in coastal sharks when compared to oceanic species [22]. If shark products are legally required to carry correct and comprehensive labelling that indicated the species and where it was caught, a consumer then has the ability to select products that come from species of sharks acknowledged to contain lower levels of mercury, purchase products caught from a sustainable fishery, or from a species that is not endangered.

In this paper, we take a mini-DNA barcoding approach to identify the shark species found in shark fin soups purchased in Singapore. The fins used in soup are already processed, meaning that any DNA is likely degraded, with this DNA becoming further degraded by heat in the cooking process. Consequently, amplifying the full 650 bp cytochrome c oxidase 1 (COI) region becomes difficult, if not impossible. To overcome these challenges, mini-DNA barcoding techniques that use a small fragment of the original gene have been developed [28,29,30]. Using this smaller fragment, we expect to identify many endangered and CITES-listed species. It is also highly probable that many of these fins will come from species that have previously been shown to contain high levels of mercury and other aquatic toxins or toxic elements.

## 2. Materials and Methods

We sequenced 92 samples collected from bowls of shark fin soup purchased at various locations throughout Singapore. Where present, intact fins were removed. When no obvious fins were present, we collected ceratotrichia (Figure 1). Fins and ceratotrichia were rinsed in sterile deionized water, cooled on ice to prevent further DNA degradation and then placed in individual sterile falcon tubes, transported to the laboratory on ice and stored at −80 °C until DNA extraction was performed.

DNA was extracted using a blood and tissue kit (Qiagen) following the manufacturer’s instructions. Where more than one sample was collected from each soup, a separate extraction was performed for each. We initially attempted to amplify an approximate 300 bp fragment of the cytochrome c oxidase subunit I (COI) gene using the mlCOIintF (5′-GGW ACW GGW TGA ACW GTW TAY CCY CC-3′) and LoboR1 (5′-TAA ACY TCW GGR TGW CCR AAR AAY CA-3′) primers [31,32] using the following Polymerase chain reaction (PCR) cycling conditions 94 °C for 60 s, 5 cycles of 94 °C for 30 s, 48 °C for 120 s, 72 °C for 60 s, and 35 cycles of 94 °C for 30 s, 54 °C for 120 s, 72 °C for 60 s and 72 °C for 5 min. Each reaction contained 1.0 μL of MgCl2 (2.5 mM), 1.0 μL of each primer at 10 μM, 1 μL BSA (20 mg/mL), 2 μL of DNA template, 12.5 μL GoTaq mastermix green (Promega), and PCR grade water to 25 μL. We then attempted to amplify any reaction that failed under the conditions described above with the following primer pair that amplifies an approximate 150 bp fragment of the COI gene. M-13 tailed forward primer VF2_tl (5′-GTA AAA CGA CGG CCA GTC AAC CAA CCA CAA AGA CAT TGG CAC-3′) and reverse primer Shark150R (5′ -AAG ATT ACA AAA GCG TGG GC-3′) [30]. Reactions were performed in 25 μL volumes, each reaction contained 12.5 μL GoTaq mastermix green (Promega), 1 μL forward primer (10 μM), 1 μL reverse primer (10 μM), 8.5 μL PCR grade water, and 2 μL of undiluted DNA template. PCR thermal cycling conditions followed an initial denaturation period of 2 min at 94 °C, followed by 35 cycles at 94 °C for 1 min, 52 °C for 1 min, 72 °C for 1 min, and a final extension period of 10 min at 72 °C.

PCR products were cleaned and Sanger sequenced by Macrogen (Seoul, Korea). Geneious v2020.2.4 (http://www.geneious.com; accessed 15 March 2022 [33] was used to view sequence data. We used The Barcode of Life Data System (BOLD, https://www.boldsystems.org; accessed 15 March 2022) and the Nucleotide BLAST (BLASTn) function in Genbank (http://www.ncbi.nlm.nih.gov; accessed 15 March 2022) to make species identifications from DNA data. Species identifications were considered positive, and unambiguous if BOLD indicated the ID was solid with no closely allied congeneric species currently known, and the same species was then identified as the top match in BLAST.

## 3. Results

We sequenced 92 samples collected from 14 bowls of soup and successfully identified 14 species of sharks at the genus or species level and 1 chimaera (Table 1). Of these, four species are listed on CITES Appendix II and tenare determined to be Critically Endangered (CR), Endangered (EN) or Vulnerable (VU) under the International Union for Conservation of Nature (IUCN) Red List of Threatened Species (Table 1). Six samples did not amplify or produce a usable sequence. If the same species was identified more than once in the same bowl of soup, we considered it the same individual.

*Prionace glauca*, the blue shark, was found in the most bowls of soup (8 of 14), *Galeorhinus galeus*, the school shark, in three and *Hemigaleus microstoma, Mustelus antarcticus,* and *Mustelus* spp. each found in two bowls; the rest only occurred once.

## 4. Discussion

Similar to other work that used DNA barcoding to identify the species of sharks involved in the fin trade [1,10,34,35,36,37], this study showed that a number of CITES Appendix II listed sharks, along with several other species of shark that are listed as threatened (Critically Endangered, Endangered, or Vulnerable) under the IUCN Red List of Threatened Species, are available to consumers in Singapore, and in this case directly for public consumption in the form of shark fin soup. From a conservation and fisheries management perspective, knowing what species are involved in the trade is essential, particularly if they are endangered [12,38,39]. Not knowing what species of sharks are involved in the trade makes designing successful management strategies and establishing correct CITES and IUCN designations very difficult. Additionally, setting appropriate catch quotas to ensure legality of products and sustainability of wild populations becomes very challenging, if not impossible when accurate identifications cannot be made [36].

Given the acknowledged role that biodiversity plays in promoting ecosystem stability and functioning [40,41,42], along with the structuring influence of large predators [40] and other apex predators can have on marine ecosystems [43], the declines in shark populations—some of which have been reduced by more than 70%—is troublesome [8], particularly as the shark product and fin trade involves many endangered species [44] and appears to continue largely unabated and unenforced.

The continuing global consumption of shark fins and products suggests consumers are either unconcerned by shark population declines, are unaware that their actions may be contributing to these declines, or do not know they could be eating endangered species. If the precarious conservation status of many sharks is not enough to discourage the consumption of shark fins and products, the high levels of elements such as mercury should be concerning to consumers, particularly as they have an established and unambiguous track record of causing severe disorders and adverse medical conditions in humans. Evidence from fins collected and analyzed throughout the world indicates that the presence of toxic elements in sharks is a global phenomenon, not restricted to specific bodies of water. Studies from the Atlantic, Indian, Pacific, Caribbean, South China Sea and Australian waters all find high levels of lead, arsenic and mercury in shark products, frequently exceeding safe advisory levels [20,21,22,23,24,26,45,46,47], and many of these tainted products enter human food chains, especially shark fins, via soups.

The most commonly encountered shark in this work was the blue shark. This shark was found in 8 of 14 bowls, and though not listed under CITES and classified as Near Threatened under the IUCN Red List of Threatened Species, blue sharks are one of the most commonly encountered sharks in the global fin trade and are traded extensively throughout Hong Kong and Singapore [9,12,48,49]. Scientific evidence suggests that this species is overexploited and should have its catch regulated to avoid population crashes [44,50]. Blue sharks collected from the Atlantic and Australian waters have been documented to contain high levels of mercury and selenium in muscle and liver tissue [25,51]. Dried blue shark fins and blue shark fins collected directly from bowls of soup in Hong Kong have levels of mercury frequently exceeding Hong Kong, European Commission and United States regulatory body maximum permissible levels [22]. While we have not determined the concentrations of mercury, or other toxic elements present in our samples collected from Singapore, it seems reasonable to suggest that the levels of mercury and other metals present in these samples will be comparable to other regions, more so given the global nature of the fin trade and the incidence of reports documenting high levels of these contaminants in blue sharks from the planet’s oceans. It is a similar story with the other sharks we identified in this paper, especially the silky and hammerhead sharks.

Anecdotal evidence collected from visiting retail establishments selling dried shark products suggests that all species of hammerhead shark fins are highly regarded and command a premium price in Singapore, which appears to be corroborated by our data. The most expensive bowl of soup was the only one that contained fins from the CITES Appendix II listed scalloped hammerhead shark (*Sphyrna lewini*). The size of the fin is also a key determinant in pricing, with larger fins commanding a higher price. While it is impossible to determine the size of the fin from the collected ceratotrichia, this bowl also contained fins from the pelagic thresher shark (*Alopias pelagicus*), a species of shark renowned for its large caudal and distinctive pectoral fins, which could also help to account for the higher price of this bowl. Soups in this study ranged in price from a minimum of USD 9.11 to a maximum of USD 53.49.

Over a quarter (29%) of the sharks we identified are listed on CITES Appendix II, and 10 of the 14 bowls contained at least one species of shark that is categorized under the IUCN Red List of Threatened Species as Critically Endangered, Endangered, or Vulnerable, making it extremely likely consumers in Singapore are eating species of sharks that are threatened with extinction and in some cases trade regulated. Additionally, and for the reasons discussed earlier, there is a high chance that consumers could be unknowingly ingesting toxic elements, such as mercury, arsenic, selenium and others, at concentrations that exceed maximum recommended levels. Mandatory, effective labelling of foods that contain shark fins and other shark products, detailing the particular species and where it was caught, would allow the consumer to make an informed choice on whether it was safe to eat and if the species was endangered or came from a potentially sustainably managed stock [11].

## 5. Conclusions

Overall, our work shows that endangered species of sharks continue to be consumed in Singapore, and it is also extremely probable that consumers of shark fin soup are exposing themselves to unsafe levels of mercury. If shark fishing continues unabated, it is very likely they will be completely removed from marine ecosystems in the not-too-distant future. Sustainable shark fishing is possible, and the better labelling of products with the species names and geographic origin would offer a way to prevent the overexploitation of species that are already endangered. Additionally, if products were labelled in a more comprehensive and rigorous fashion, the consumer would be able to make a more informed choice and only purchase fins and other shark products from fisheries that are acknowledged to be sustainable. Better labelling could also be important from a human health perspective; for example, if certain species of shark are known to accumulate mercury at higher concentrations, then these could then be avoided.

## Figures and Tables

**Figure 1 animals-12-00802-f001:**
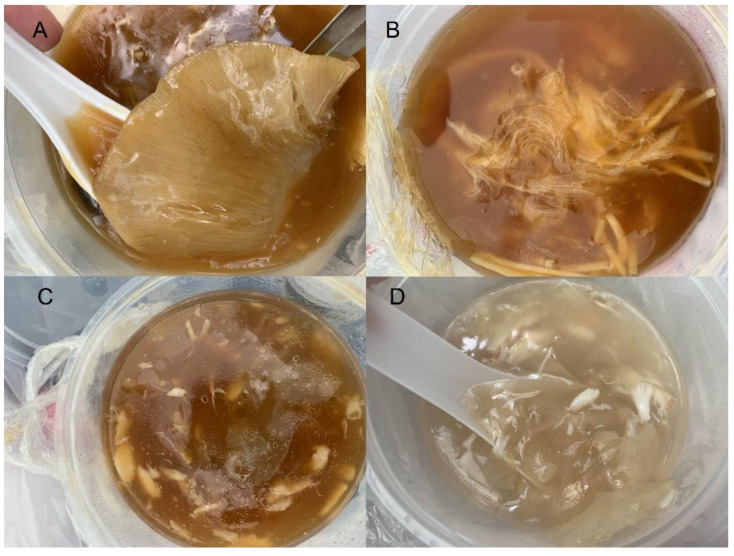
Examples of the products purchased in this work. (**A**) = soup 1, (**B**) = soup 2, (**C**) = soup 3 and (**D**) = soup 4. See Table 1 for the species identified in each.

**Table 1 animals-12-00802-t001:** Details of the species of sharks in shark fin soups collected in Singapore, their common names and conservation status according to the International Union for Conservation of Nature (IUCN) Red List of Threatened Species and Convention on International Trade in Endangered Species of Wild Fauna and Flora (CITES) listings.

	Price USD	Number of Samples Taken	Number of Species Identified	Species Identified	Common Name	CITES Listing	IUCN Red List Status
Soup 1	15.66	5	2	*Prionace glauca*	Blue shark	--	NT
				*Carcharhinus falciformis*	Silky shark	II	VU
Soup 2	53.49	10	2	*Alopias pelagicus*	Pelagic thresher	II	EN
				*Sphyrna lewini*	Scalloped hammerhead	II	CR
Soup 3	20.59	5	3	*Prionace glauca **	Blue shark	--	NT
				*Hemigaleus microstoma **	Sicklefin weasel shark	--	VU
				*Carcharias taurus **	Sand tiger shark	--	CR
Soup 4	9.11	5	1	*Galeorhinus galeus*	School shark	--	CR
Soup 5	11.28	5	2	*Prionace glauca*	Blue shark	--	NT
				*Rhizoprionodon acutus*	Milk shark	--	VU
Soup 6	29.93	5	3	*Mustelus antarcticus*	Gummy shark	--	LC
				*Galeorhinus galeus*	School shark	--	CR
				*Squalus spp*	--	--	--
Soup 7	45.63	5	3	*Hemigaleus microstoma*	Sicklefin weasel shark	--	VU
				*Prionace glauca*	Blue shark	--	NT
				*Carcharias taurus*	Sand tiger shark	--	CR
Soup 8	36.62	6	1	*Prionace glauca*	Blue shark	--	NT
Soup 9	22.87	6	2	*Mustelus henlei*	Brown smooth-hound	--	LC
				*Mustelus spp **	--	--	--
Soup 10	30.88	12	2	*Prionace glauca*	Blue shark	--	NT
				*Alopias superciliosus*	Bigeye thresher	II	VU
Soup 11	14.15	5	1	*Loxodon macrorhinus*	Sliteye shark	--	NT
Soup 12	33.09	10	2	*Prionace glauca*	Blue shark	--	NT
				*Callorhinchus callorynchus*	American Elephantfish	--	VU
Soup 13	41.07	6	1	*Prionace glauca*	Blue shark	--	NT
Soup 14	31.51	7	2	*Galeorhinus galeus **	School shark	--	CR
				*Mustelus spp **	--	--	--

* Indicates 150 bp fragment used for identification; all other identifications were made using an approximate 300 bp fragment.

## Data Availability

Not applicable.

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
