# Peer review of "What Is in Your Shark Fin Soup? Probably an Endangered Shark Species and a Bit of Mercury"

_animals, 2022, doi:10.3390/ani12070802_

Round 1

Reviewer 1 Report

The authors have put together a simple but very clever study in which they investigate the species of sharks found in Asian shark fin soup via detection with molecular sequencing. The results show several species of shark were found in the soups, and provide an interesting study which should be repeated in other areas.  The paper is well written and likely of interest to a wide audience. There are several points which should be explained further, which will help to enhance the conclusions which are too short for publication, and give a greater overall impact to this piece.

Major additions/issues:

A photo of shark fin soup from this study must be included as a figure. This will enhance interest in the reader, and help to show how difficult ID would be without sequencing.

The cost of the soup should be discussed more (in USD$).

The taste of the fin is apparently nothing (but cite this), but what is the shark soup flavor overall?

The MS has several interesting points, but falls short to capatilize on the plight of the shark species, and the combine negative effects of human health. Discussing how removing an apex predator from the oceans is also extremely negative would enhance the MS (a plethora of literature is available on this, such as system collapse). Discuss the conservation end, what is the way forward to save sharks?

Are there other examples of such a study? these should be discussed within the text. 

Blue sharks, which are not critically endangered were the most encountered in this study. It is possible this is because they are the most common currently in hunting areas, and thus the most commonly caught. With this in mind, it should be discussed that continued demand for fins will likely cause a population crash within this species.

L150 talks about a 150 or 300bp sequence used for identification. Without seeing these sequences deposited, or any data on them, this is questionable. Are these values standard for shark ID? Discuss in the text either way.  Since sequences were created for this study, why were they not deposited in Genbank for future researchers to utilize in their studies? They should be listed in full with accession numbers, or a strong argument for their omission should be discussed. Since these are not presented in this draft, the results and species ID are questionable as they cannot be examined. I do not suggest an entire phylogenetic tree be constructed for this study, but the sequences should be made available.

The conclusions are only 3 lines. This is far to short and must be amended in the next version. There are many interesting points to discuss from the MS and that should be added. The conclusion should talk about the potential issues with global shark populations, fisheries, protection of species, human health etc etc. As is stands, it seems as if this section is missing. 

Minor edits:

L 22. Add ‘shark’ before declines.

L 26 is gelatinous a better descriptor than “fibrous” ?

L 37 add a bit more here about the decline of shark species as a result of continued shark fin soup consumption.

L 41 add ‘is considered’ before delicacy to not promote it. The fins are also apparently tasteless, and have no real medical benefit (e.g. anti cancer, sexual enhancement). This could be mentioned here- as you show later consuming them can be deleterious to human health.

L 56 mention how many sharks are finned and thrown overboard (so as not to take up weight on the ship), therefore increasing the take of such illegal practices.

L64 switch ‘microbes’ to bacteria (the primary driver of this transfer)

L 73 what is meant by ‘risk assessments’ ?

L 91 remove ‘with this work’ and reword

L 92 ‘we expect’ was this a hypothesis? It seems out of place here.

L 102. What type of water were they rinsed in? sterile DI water, or bottled etc?

L 103 journal to fix the degrees symbol.

L 114 what type of water? And throughout such as L121

L122 symbol looks odd. And L123 also.

L 131- were sequences uploaded to databases such as Genbank? If so add excision numbers etc. Data missing are the % matches.

L 141. ‘school’? or species.

L 151 the price- please take these values out and convert to USD$ and discuss in the text, showing the range etc.

L 176 why the sudden mention of arsenic? There are other troubling compounds also within sharks, so mention all in previous sections or only the one.

L197 A good point that a fin trade is global, so it is unknown where a given fin may come from, so toxins are even more difficult to avoid for a consumer.

L203 as above, discuss the prices in USD$. Shark fin soup has been reported as a status symbol (due to its price presumably) so discuss that further.

L 211 ‘categorised’ is the British spelling. Check throughout MS to either keep it all UK, or switch all to USA should the journal have such requirements.

L227-229 is this section needed? ‘all authors contributed equally’ or adding nothing would be sufficient.

L 233- what grant number, and where is the fund located?

L238-240 delete template text

Reviewer 2 Report

Using DNA barcoding technique, Pei Choy and Wainwright identified the presence of endangered shark species in fin soups purchased in Singapore. Given the steep decline in shark populations all over the world and the rise in demanding for shark products, this manuscript surely contribute to shed some light on the presence of protected species involved in the Asian fin trade. Therefore, this manuscript provides important information to the scientific community, and thus worthy for publication after a minor revision, please see specific comments below:

- Title: I would suggest to modify as follows: “What’s in your shark fin soup? Probably an endangered shark species and a bit of mercury.”

- Abstract:

Line 28-30: maybe the authors can rephrase as: “The most common species identified in our samples was the blue shark (Prionace glauca), a species listed as Near Threatened on the International Union for Conservation of Nature (IUCN) Red List with a decreasing population, on which scientific data suggests catch limits should be imposed.”

Line 32: “We identified four other shark species that are listed on CITES Appendix II, and in total nine species that are assessed as Critically Endangered, Endangered or Vulnerable under the IUCN Red List of Threatened Species”

- Introduction:

Line 42 – the manuscript by Ip et al. 2021 (e.i. reference 2) focuses on the use of eDNA metabarcoding on seawater samples to detect elasmobranch species in the hyper-urbanised waters off Singapore, therefore it is not an appropriate reference in this context. The authors should cite a different study.

Line 58 – the reference should be formatted on a number basis.

Line 61 – check out the reference format

Line 62 – the abbreviation “Hg” should be used in line 57, when the authors deal with mercury for the first time

Line 61-64 – the authors should rephrase this line

Line 79 – nice, indeed a recent study analyzed the accumulation of 12 different trace elements in multiple tissues (fin, muscle, and liver) from 12 elasmobranch species and showed not only biomagnification of Hg in the shark assemblage but also that the cartilage (fin) tissue showed a different accumulation profile compared to muscle and liver with Pb, Cr and Co occurring in the highest concentrations. It might be worthy to be added in the manuscript:

Boldrocchi, G., Spanu, D., Mazzoni, M., Omar, M., Baneschi, I., Boschi, C., ... & Monticelli, D. (2021). Bioaccumulation and biomagnification in elasmobranchs: A concurrent assessment of trophic transfer of trace elements in 12 species from the Indian Ocean. Marine Pollution Bulletin, 172, 112853.

Line 86 – same comment to line 79

Line 101 – ceratotrichia was collected.

Line 103 –  -80 °C

Line 134 – Chimaera should be in italics

Line 155 – instead of “we show that” maybe rephrase as “this study showed that”

Line 181-183 – Nice, at this point I would suggest the authors to add the remaining basin: Indian Ocean, for instance:

Le Bourg, B., Kiszka, J. J., Bustamante, P., Heithaus, M. R., Jaquemet, S., & Humber, F. (2019). Effect of body length, trophic position and habitat use on mercury concentrations of sharks from contrasted ecosystems in the southwestern Indian Ocean. Environmental research, 169, 387-395.

Boldrocchi, G., Monticelli, D., Omar, Y. M., & Bettinetti, R. (2019). Trace elements and POPs in two commercial shark species from Djibouti: Implications for human exposure. Science of the Total Environment669, 637-648.

Line 185 – no need for the Latin name here as the authors have already used it in line 140

Line 199-200 – the authors should rephrase these lines

Line 201-203 - the authors should rephrase these lines, moreover: hammerhead shark? Which species? If I am not wrong the authors are referring to the scalloped hammerhead shark, so please write the full common name.

Line 204 – if authors referred to the scalloped hammerhead shark in line 201-203, please add the Latin name there and remove it from line 204.

Line 201-209 – I suggest the authors to either remove this part or rephrase it

References: all the scientific names (e.i Latin name) of species should be italicized, please see reference 15, 17, 34… etc

Round 2

Reviewer 1 Report

The authors of this fintastic study have done well to revise their MS into an excellent paper. I have a few remaining minor suggestions below which I hope will enhance their text. This work will make a significant positive contribution to the journal Animals:

The author response “It is necessary to use these mini-barcodes because the DNA in the soup is degraded by the cooking process, it is not possible to get the full length barcode to amplify.”     This well said, and I feel would enhance the MS by its inclusion. Scientists from other fields will certainty read this piece, and this clarity would enhance their (as it did to my) understanding of the particular methods- which strengthens their study considerably. Someone who deals with microbes reading this piece is used to seeing entire genes or genomes, so adding a line about the rationale behind this data set of sequences is useful.

 Author responds: “To our knowledge shark fins are generally not sold or consumed for medical purposes, rather, they are consumed as a luxury product to show wealth/status usually at celebratory events (e.g., weddings)”

According to Wikipedia (fully acknowledging the dubiousness of this) “In traditional Chinese medicine, shark fins are believed to help in areas of rejuvenation, appetite enhancement, and blood nourishment and to be beneficial to vital energy, kidneys, lungs, bones, and many other parts of the body”. Although their citation there has expired, a google search of ‘shark fin soup health benefits’ yields 676,000 results. There is a large public perception that Asian markets sell this soup (along with status) as a medical benefit (similar to their consuming various large African mammal horns…). PADI educates millions of divers on this subject, and suggests the perceived (obviously falsely) health benefits are a driving factor (e.g. https://blog.padi.com/how-not-to-be-wrong-about-sharks/). This makes your argument in the text that eating shark fins is very negative for human health all the more poignant. I am sure that there are several peer reviewed publications which mention this health idea as a market factor, and they should be cited as this is also clearly the public perception. I suggest adding 1 or two lines about this with a citation or two.

A question I now have after seeing (the excellent) added figures of the soup – is it served hot or cold? Might be interesting to slip that in somewhere.
